# Pseudophosphatases as Regulators of MAPK Signaling

**DOI:** 10.3390/ijms222212595

**Published:** 2021-11-22

**Authors:** Emma Marie Wilber Hepworth, Shantá D. Hinton

**Affiliations:** Integrated Science Center, Department of Biology, College of William and Mary, Williamsburg, VA 23185, USA; ewhepworth@email.wm.edu

**Keywords:** mitogen-activated protein kinase (MAPK), extracellular signal-regulated kinase (ERK), kinase, protein tyrosine phosphatases (PTPs), pseudophosphatase, dual-specificity phosphatases (DUSPs), MAPK phosphatase (MKP), phosphoserine/threonine/tyrosine-interacting protein (STYX), MAPK phosphoserine/threonine/tyrosine-binding protein (MK-STYX), TAK1-binding protein (TAB 1)

## Abstract

Mitogen-activated protein kinase (MAPK) signaling pathways are highly conserved regulators of eukaryotic cell function. These enzymes regulate many biological processes, including the cell cycle, apoptosis, differentiation, protein biosynthesis, and oncogenesis; therefore, tight control of the activity of MAPK is critical. Kinases and phosphatases are well established as MAPK activators and inhibitors, respectively. Kinases phosphorylate MAPKs, initiating and controlling the amplitude of the activation. In contrast, MAPK phosphatases (MKPs) dephosphorylate MAPKs, downregulating and controlling the duration of the signal. In addition, within the past decade, pseudoenzymes of these two families, pseudokinases and pseudophosphatases, have emerged as bona fide signaling regulators. This review discusses the role of pseudophosphatases in MAPK signaling, highlighting the function of phosphoserine/threonine/tyrosine-interacting protein (STYX) and TAK1-binding protein (TAB 1) in regulating MAPKs. Finally, a new paradigm is considered for this well-studied cellular pathway, and signal transduction pathways in general.

## 1. Introduction

Signal transduction is an intricate dance of cells sensing and responding to their environment, as well as influencing the behavior of neighboring cells or an entire organism [1]. These complex intercellular and intracellular signaling networks define the cellular milieu. Post-translational modifications (PTMs) of proteins also play a role in the complexity of signaling networks. PTMs are reversible or irreversible chemical processes that modify one or more amino-acid residues of proteins after they are translated [2]. Eukaryotic PTMs include acetylation, biotinylation, formylation, glycation, glycosylation, hydroxylation, methylation, oxidation, palmitoylation, nitrosylation, SUMOylation, ubiquitination, and phosphorylation [3].

Protein phosphorylation is the most common PTM, regulating many diverse cellular responses such as enzymatic activity, protein–protein interactions, and protein intracellular localization, half-life, and turnover [2]. Kinases and phosphatases regulate phosphorylation; kinases add a phosphate group (PO_4_^2−^) to a substrate, while phosphatases remove the attached phosphate. The most common phosphorylation events occur on serine, threonine, and tyrosine residues [4]. At least thirty percent of the human proteome can potentially be modified by approximately 568 kinases to regulate the majority of cellular pathways [5]. However, both kinases and phosphatases are required to control the activity of target proteins by coordinating a fine balance between phosphorylation and dephosphorylation. For example, receptor tyrosine kinases (RTKs) dimerize when bound to a ligand, triggering transphosphorylation of the RTK cytoplasmic domains, and activating multiple downstream pathways such as the phosphatidyl 3-kinase/AKT pathway [6] and the mitogen-activated protein kinase/extracellular signal-regulated kinase (MAPK/ERK) pathway [7].

### 1.1. Mitogen-Activated Protein Kinase (MAPK) Signaling

Mitogen-activated protein kinase (MAPK) signaling pathways are important regulators in cellular processes such as survival, proliferation, apoptosis, and differentiation [8]. In addition to cytoplasmic functions, MAPKs regulate some cellular processes after translocation to the nucleus. Within the nucleus, MAPKs modulate gene expression by phosphorylating and activating transcription factors [9,10]. There are multiple mammalian MAPK pathways, including extracellular signal-regulated (ERK1/2); c-JUN NH_2_-terminus kinase or stress protein-activated kinases (JNKs/SAPKs); and p38 MAPK [9]. ERKs are mostly activated by insulin, mitogens, and growth factors, whereas JNK and p38 MAPK are primarily activated through environmental stresses and proinflammatory stimuli [8,9,11,12]. Thus, MAPKs also have essential roles in stress- and inflammation-activated pathways, linking them to diabetes, inflammatory disorders, and cancer [8,13].

The existence of a three-tiered “core signaling module,” consisting of MAP kinase (MAPK), MAP kinase kinase (MAP2K) and MAP kinase kinase kinase (MAP3K), highlights the importance of the tight control required for activation of MAPK pathways. The core signaling module elicits phosphorylation and activation simultaneously through the conserved Thr-X-Tyr motif within the kinase activation loop; phosphorylation of both residues is required for high activity [9]. Thr-X-Tyr phosphorylation is catalyzed by MAPK kinases (MAP2Ks) also known as MEKs or MKKs, which are activated by Ser/Thr phosphorylation within the activation loop [10,13,14]. A diverse family of kinases termed MAP3Ks catalyzes this phosphorylation [13]. The phosphorylation of MAPK is transient and reversible [9]; it is dephosphorylated by dual-specificity phosphatases (DUSPs) or MAPK phosphatases (MKPs) [9,15,16,17], demonstrating that MKPs are also vital for proper function of MAPK pathways.

It is apparent that kinases and phosphatases have critical roles in regulating MAPK signaling pathways. Attention is now being paid to the catalytically inactive members of these superfamilies, pseudokinases and pseudophosphatases, leading to reports that these pseudoenzymes are also important signaling regulators [18,19,20,21]. Certain pseudophosphatases have been shown to be regulators of MAPK signaling, highlighting the timeliness of this special edition, *Mitogen-Activated Kinases: New Insights for Old Cell Signaling Pathways*. This review discusses two pseudophosphatases with divergent roles: MAPK phosphoserine/threonine/tyrosine-binding protein (MK-STYX), a member of the MKP subfamily [15,16,17,20,22], does not directly regulate ERK/MAPK signaling [22,23], whereas the non-MKP pseudophosphatase phosphoserine/threonine/tyrosine-interacting protein (STYX) [24] does directly regulate ERK MAPK signaling [25,26]. The regulation of ERK/MAPK by SYTX, and TGF-beta-activated kinase 1 (TAK1)-binding protein (TAB1) activation of p38 MAPK [27], indeed provide new insight into the well-studied MAPK signaling pathways.

### 1.2. Pseudophosphatases as Signaling Molecules

The pseudoenzyme superfamily, first revealed through genomics, consists of a prevalent family of pseudophosphatases, which lacks the ability to catalyze removal of phosphorylated residues [17,20,24,25,28,29,30]. There is an estimated 13.8% of pseudophosphatases within the human phosphatome. Their enzymatic inactivity is due to mutations of critical residues important for the formation of the catalytic center [18,19,20,31]; however, many of these pseudoenzymes maintain a three-dimensional fold [17,18,20], allowing them to bind phosphorylated proteins to regulate other cellular processes [17,20,21,25,32]. For example, pseudophosphatases play a role in spermatogenesis, stress response, apoptosis, neuronal differentiation, cell fate, migration, ubiquitylation, demyelination, and transcription [20,21,22,25,28,32,33,34,35,36]. Within the past decade, pseudophosphatases have emerged as key regulators in signal transduction cascades by serving as competitors, signaling integrators, modulators, and anchors in cellular processes (Figure 1) [20,21,25].

Moreover, misregulation of pseudoenzymes has been implicated in various diseases such as cancer and neurological disorders [20,21,25,28,37,38]. As an example, myotubularins (MTMs) are the most prevalent of the pseudophosphatases, forming pseudophosphatase:phosphatase heterodimer signaling complexes. These heterodimer signaling complexes are required for cellular processes such as differentiation, membrane trafficking, endocytosis, and survival [20,29,39,40,41]. Moreover, mutations in the pseudophosphatase or phosphatase components, as well as disassociation of the complex can lead to disease [20,28,42,43], such as Type 4B Charcot–Marie–Tooth disease (abnormal nerve myelination) [28,39,44] and axon degeneration [42,44]. In addition, pseudophosphatases have been implicated as regulators in MAPK signaling pathways, the focus of this review.

## 2. MAPK Phosphatases (MKPs) Role in MAPK Signaling

Classical MKPs are DUSPs that possess two domains, an *N*-terminal CH2 (CDC25 [cell division cycle 25]/rhodanese homology) non-catalytic domain and a C-terminal catalytic phosphatase domain (Figure 2A) [45,46,47]. These domains are important for MKPs to dephosphorylate MAPKs (Figure 2B); both threonine/serine and tyrosine residues within the Thr-X-Tyr activation loop of MAPKs are dephosphorylated by MKPs [48,49,50]. The dephosphorylation of MAPKs is achieved through the highly conserved C-terminal domain which has the HCX_5_R active site signature motif, which is specific for a tyrosine phosphatase [51,52]. Furthermore, this HCX_5_R domain is extended to also achieve dephosphorylation of threonine and serine residues [29,53]. The non-catalytic *N*-terminal CH2 domain is more divergent in MKPs, and plays a role in substrate specificity [9]. The MAPK substrate is positioned for effective catalysis (Figure 2B) through the kinase-interacting domain (KIM) within in the CH2 domain [15,22,47]. MKPs may regulate multiple MAPK pathways. For example, the prototypical mammalian DUSP-1/MKP-1 dephosphorylates ERKs, c-Jun amino terminus kinase, and p38 [50,54], illustrating the complicated regulation of MAPKs by MKPs. A notable exception to MKP family members regulating MAPKs, however, is the pseudophosphatase MK-STYX.

### Atypical MKP: MK-STYX

Of the eleven members of the MKP subfamily, MK-STYX is the only catalytically inactive member [15,16,20,22,24,45]. MK-STYX is also known as DUSP-24 or STYXL1 [16,22]. It lacks the critical cysteine in its active site signature motif (HCX_5_R), which is essential for phosphatase activity (Figure 3A) [16,17,20,22,55]. Because MK-STYX is a member of the MKP subfamily, it is reasonable to think that it would have a role in MAPK signaling. MK-STYX is homologous to DUSP-1/MKP-1 and DUSP-6/MKP-3 [15,16,20,22,24,45]; however, MK-STYX does not regulate ERK/MAPK phosphorylation [20,22,23,56,57]. Further detailed bioinformatic analysis of MK-STYX demonstrates that it also has mutations within the KIM motif of the CH2 domain (Figure 3B) [22]. Consecutive arginine residues, which MK-STYX lacks (Figure 3B) [22], are critical within the KIM domain for MAPK/ERK docking [50,58]. Perhaps, these mutations are why MK-STYX does not bind MAPK/ERK and does not impact MAPK/ERK1/2 signaling (unpublished data and Niemi et al.) [22,23].

The KIM of active MKPs is closely linked to substrate specificity and binding of the target MAPK, inducing proper activation of the MKP. Because the *N*-terminal KIM binds a substrate independently of the C-terminal active site, a mutation within this motif has the potential to affect a pseudophosphatase such as MK-STYX. When the predicted macromolecular structure of MK-STYX was generated, refined, validated, and mutated to restore the consecutive arginines, similar to those within the sequence of MKP-3, there was a conformational change (Figure 3). Intriguingly, these bioinformatic approaches show that unlike the active site mutation, the KIM mutation does in fact affect the overall structure of MK-STYX. PyMOL, Pocket Cavity Search Application (POCASA) and Missense3D analyses of the KIM motif of MK-STYX predicts that the lack of arginine changes the expected shape of the KIM-binding pocket of the protein (Figure 3C) [63,64,65]. The KIM motif (blue area) (^48^ITALR**V**KKKN^57^) of wild-type (WT) MK-STYX is larger than the mutant (black arrow), which substituted the arginine and restored the KIM motif, ^48^ITALRRKKKN^57^ (WT residue shown in yellow and mutated residue shown in red) (Figure 3C). This change in shape and decrease in volume of the binding pocket of the mutated MK-STYX_V53R indicates that MK-STYX may bind a unique set of substrates different from its active MKP homologs. Furthermore, MK-STYX has been reported to bind non-MAPK proteins such as G3BP-1 (Ras-GTPase-activating protein SH3 domain-binding protein-1) [16,20,25] and PTPM1 (PTP localized to the mitochondrion 1) [20,25,67]. MK-STYX-G3BP-1 interaction implicates MK-STYX in the stress response pathway [16,68], whereas its interaction with PTPM1 implicates it in the apoptotic pathway [23,67]. In addition, MK-STYX is a signaling molecule in neurite formation pathway [20,56,57,69]. These studies used PC12 cells, which requires sustained ERK/MAPK activation for neurite formation [56,70]. However, MK-STYX did not sustain ERK activation in these cells, but modulated the RhoA signaling pathway [20,56]. This further illustrates that MK-STYX has specific binding partners beyond the MAPKs.

## 3. STYX

The prototypical pseudophosphatase STYX [17,24], which is not a member of the MKP family, but a DUSP [17,24], reduces MAPK activation by binding ERK1/2 [25,26,71]. STYX has a glycine residue in place of the essential active-site cysteine [24] (Figure 4A), and a single point mutation of glycine to cysteine restores its catalytic activity [24]. The role of STYX in cell fate and migration revealed its function as a regulator of the ERK1/2 MAPK signaling pathway. STYX inhibits differentiation by competing with MKP-2 (MAPK phosphatase-2) for binding to ERK1/2 (Figure 4B), preventing ERK1/2 from exiting the nucleus (Figure 4B) [21,22,25]. STYX is capable of binding both the phosphorylated and unphosphorylated forms of ERK [26,71]. Therefore, STYX modulates cell-fate decisions and cell migration through spatiotemporal regulation of ERK1/2 signaling by serving as a competitor in a dominant negative role (Figure 4B) [20,21,25]. STYX also affects directional cell migration by disrupting the morphology of the Golgi apparatus in an ERK-dependent manner [20,25,26,72]. STYX also plays a role in the ubiquitination pathway, by regulating the ubiquitin ligase SKP/CUL1-F-box (SCF) complex [20,22,25,72], and by interacting with the F-box protein WD40 domain (tryptophan and aspartic acid repeats), FBXW7 [20,25,26]. FBXW7 is a substrate recruiter for a ubiquitin protein ligase complex [72], and a tumor suppressor [20,25,73,74,75]. This interaction typically occurs in a phosphorylation-dependent manner, linking ubiquitination and phosphorylation [20,72]. STYX’s interaction with ERK1/2 and FBXW7 solidifies its role in MAPK signaling and the ubiquitination pathways, further supporting that these less studied proteins are new players to consider as regulators of these “old” cellular pathways such as MAPK signaling.

## 4. TAB1

Intriguingly, a serine/threonine pseudophosphatase, TAB1 [TAK1 (TGF-beta-activated kinase 1)-binding protein], which has a fold similar to MPP (Mg^2+^-dependent protein phosphatase) [76], also has a role in linking the phosphorylation and ubiquitination cascades [77]. There are three TAK-binding proteins, TAB1, TAB2, and TAB3; TAB1 constitutively binds TAK1 at its *N*-terminus and serves as a phosphorylation-dependent regulatory subunit, whereas TAB2 and TAB3 bind the C-terminus in a context-dependent manner [78]. TAK1 is a MAP3K and an upstream effector of interleukin 1 and stressor pathways. TAB1 has a downstream role, as a regulator of p38α MAPK [27,77,79]. Studies in knockout mouse embryonic fibroblasts (MEFs) demonstrate that TAB1 has an important role in recruiting p38 MAPK to the TAK1 complex, which is required for activation of TAK1 [79]. Additionally, p38 MAPK can downregulate the activity of TAK1 by phosphorylating the Ser423 and Thr431 residues of TAB1 and thereby creating a feedback-control mechanism to regulate TAK1 activity [76,80]. TAB1 has roles in other pathways; it blocks inhibition of p53 [81] and has been implicated in ovarian cancer [81] and pro-inflammatory diseases such as rheumatoid arthritis [77,78,79]. TAB1 actions within the TAK1 signaling pathway exemplify how the “grab and hold on” [32] technique of pseudophosphatases can initiate and serve many mechanistic modes of actions such as integrators, competitors, anchors, and modulators in signaling pathways [21,25].

## 5. Conclusions

Reviewing the contribution of pseudophosphatases in MAPK signaling is ideal for this special edition, *Mitogen-Activated Kinases: New Insights for Old Cell Signaling Pathways*. Once perceived to be “dead” and molecular “duds” [25,31,82], they are now known to be essential signaling regulators. Studies within the dual-specificity tyrosine-phosphorylated and-regulated kinase (DYRK) provided the initial and pivotal evidence of pseudophosphatases as regulators of signaling pathways [22,32]. For example, *C. elegans* pseudophosphatases EGG4 and EGG5 trap mini brain kinase 2 (MBK-2) to regulate the oocyte-to-zygote transition though DYRK [35,36]. In this case, EGG4 and EGG5 prevent kinase binding by serving as competitors. They bind the YTY motif (activation loop) of MBK-2 and inhibit its activity, bypassing tyrosine dephosphorylation [36], demonstrating that pseudophosphatases can exert effects similar to phosphatases by inhibiting kinases [22,32,35,36].

Continued study of pseudophosphatases has provided new insights into cell signaling pathways, while shifting the paradigm for better understanding of the complexity of signaling cascades. The roles of STYX and TAB1 in MAPK signaling demonstrate that pseudophosphatases must be considered when investigating the tight control of MAPK activation and downstream regulation. Pseudophosphatases are not limited to the PTP family but are found among the serine/threonine phosphatases as well [25,83,84]. Likewise, MAPK signaling is regulated by both DUSP pseudophosphatases and serine/threonine phosphatases. These pseudophosphatases regulate various MAPK signaling pathways, including the ERK1/2 and p38 pathways.

Understanding the structure–function relationship of pseudophosphatases is important for understanding their function [22,25]. Mutations within the active enzymes that resulted in pseudophosphatases are essential and critical for their current function (Table 1). There are active phosphatases whose non-catalytic activity is critical for regulation of signaling pathway [25,28]. For example, the non-catalytic activity of phosphatase of regenerating liver-3 promotes metastatic growth [28,85].

The combination of proteomics, biochemistry, structural biology, and transgenic animal models has been instrumental in understanding the molecular mechanisms of pseudophosphatases [20,23,67]. Pseudophosphatases have emerged as bona fide signaling regulators, modulating various cellular processes such as spermatogenesis, stress response, apoptosis, neuronal differentiation, cell fate, migration, ubiquitylation, and demyelination [20,21,22,28,32,33,34,35,36]. Thus, they have been implicated in various diseases (Table 1) [28]. MAPK signaling pathways also have been implicated in many diseases such as oncogenesis, diabetes, and rheumatoid arthritis [9,13]. It is not coincidental that pseudophosphatases are important players as regulators of MAPK signaling. The continued development of new technologies and the openness to investigate “old” signaling pathways with a new perspective will lead to innovative, exciting, and profound discoveries. It is apparent that pseudoenzymes are pivotal players in signal transduction. Understanding signaling pathways may be at the crossroads, where pseudoenzymes are the connecting bridges.

## Figures and Tables

**Figure 1 ijms-22-12595-f001:**
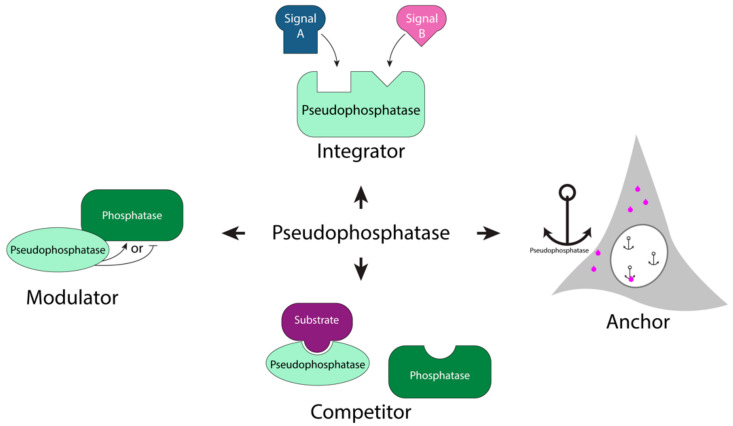
Roles of pseudophosphatases as signaling molecules. There are several different mechanisms that pseudophosphatases utilize to regulate signaling pathways. They may serve as an integrator and integrate multiple signaling pathways (top); a molecular anchor by binding a target protein and restricting its subcellular localization (right); a competitor for substrate binding (bottom); and as a modulator of enzymes such as an active phosphatase by either promoting or inhibiting the activity of the phosphatase (left) (adapted image) [20,21,25].

**Figure 2 ijms-22-12595-f002:**
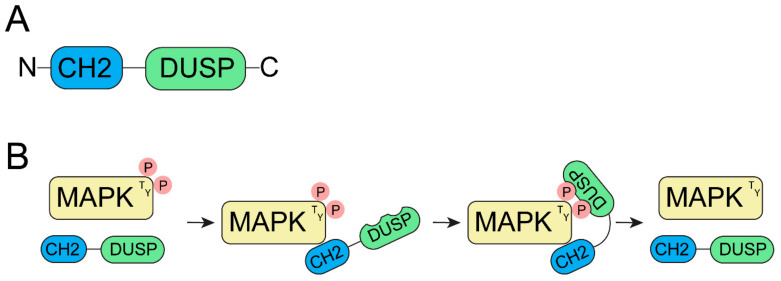
MKPs regulate MAPKs. (**A**) The domains of a canonical MKP. MKPs have two domains, a C-terminal catalytic DUSP domain that dephosphorylates MAPKs and an *N*-terminal CH2 domain that binds the MAPK-binding domain. The CH2 domain recognizes and docks the specific MAPK substrate, stabilizing it for dephosphorylation. (**B**) The DUSP domain of MKPs dephosphorylates T and Y residues in the MAPK activation loop (pTXpY), downregulating MAPK activation. The CH2 domain of the MKP binds the MAPK, inducing a conformational change, which activates and increases the catalytic activity of the MKP DUSP domain. This promotes the DUSP active site to dephosphorylate the MAPK residues, thereby inactivating the MAPK.

**Figure 3 ijms-22-12595-f003:**
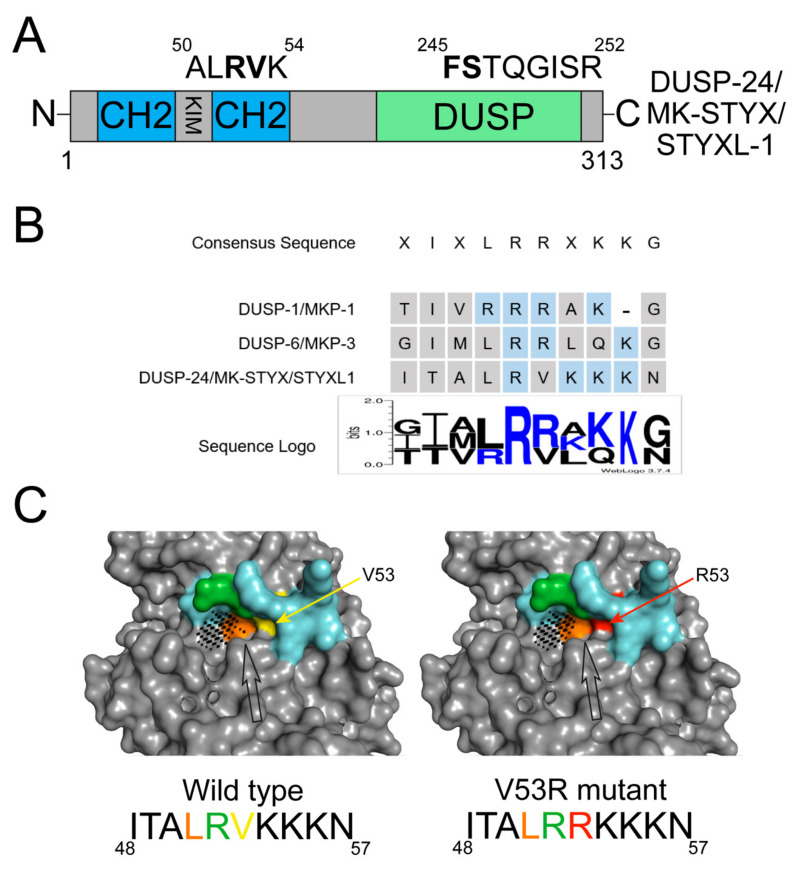
Bioinformatics structural analysis of pseudophosphatase MK-STYX. (**A**) The atypical domains of MK-STYX. MK-STYX has a C-terminal DUSP domain and an *N*-terminal CH2 domain, similar to its active homologs. A mutation in the active site of the DUSP domain from HCX_5_R to FSX_5_R renders MK-STYX catalytically inactive. In addition, the presumed kinase-interacting motif (KIM) within the CH2 domain of MK-STYX lacks consecutive arginine residues, required for MAPK binding. Therefore, MK-STYX may not bind MAPKs. (**B**) The KIM analysis of MK-STYX and active homologs, MKP-1 and MKP-3. The KIM regulates substrate specificity and substrate docking in CH2 domain of classical MKPs. The KIM requires 2 to 3 consecutive arginine residues to create a consecutive positively charged region that interacts with the negatively charged aspartic acid residues on the target MAPK substrate for docking of the MAPK. Unlike its active homologs, MKP-1 and MKP-3, the KIM consensus motif of MK-STYX contains only a single arginine. This may explain why MK-STYX does not bind and regulate MAPKs in the same manner as typical MKPs. The sequence logo was made using WebLogo 3.7.4 [59] and has a 2.0 bit scale (positive residues are shown in blue, neutral residues are shown in black, no negative residues are present). (**C**) Computational mutagenesis of the KIM of MK-STYX affects the size and shape of a predicted binding pocket. To probe for possible explanations behind why MK-STYX lacks the ability to regulate MAPK signaling, a predictive model of the macromolecular structure of MK-STYX was generated by Iterative Threading ASSEmbly Refinement (I-TASSER) [60]. The best I-TASSER model (validated through MolProbity [61] with a score of 3.03) was refined using DeepRefiner [62]. The best quality refined model (predicted global quality score of 0.142 and MolProbity score of 2.14) was mutated using the mutagenesis function in the PyMOL Molecular Graphics System 4.6.0 [63] to create the V53R mutant, which restores the consecutive arginine residues in the KIM. The structures of both wild-type (WT) MK-STYX and the V53R mutant were submitted to Pocket Cavity Search Application (POCASA) [64] to probe the surface of the protein and predict possible binding sites (1.0 Å grid size and 2 Å probe radius). When the predicted pockets were compared between the WT MK-STYX and the V53R mutant, the V53R mutant displayed a smaller and differently shaped predicted binding pocket in the area of the KIM (rank 3 out of 13 predicted pockets, decrease in volume from 98 to 96, and decrease in volume depth value of 271 to 261). The area of change (shown by black arrow) was in the immediate area of the mutated residue (WT residue shown in yellow and V53R mutation in red). In addition, the WT model was submitted to Missense3D [65], which also predicted that the V53R mutation altered a cavity and leads to the contraction of cavity volume (predicted volume contraction of 89.424 Å^3^). For further validation, the WT model was submitted again, to predict the effect of an S246C mutation in the signature active site motif; there was no predicted structural damage (data not shown). This was expected because most pseudophosphatases maintain their three-dimensional fold. This may indicate that while MK-STYX does not bind MAPKs at this site; this pocket may allow MK-STYX to bind a novel set of binding partners. Techniques used to produce the data for this study were learned through the Malate Dehydrogenase CUREs Community Workshop on Virtual CUREs and UREs (https://mdh-cures-community.squarespace.com/virtual-cures-and-ures (accessed on 27 June 2021)), supported by a grant from the National Science Foundation. NSF-1726932 EHR-IUSE, Principal Investigator Ellis Bell [66].

**Figure 4 ijms-22-12595-f004:**
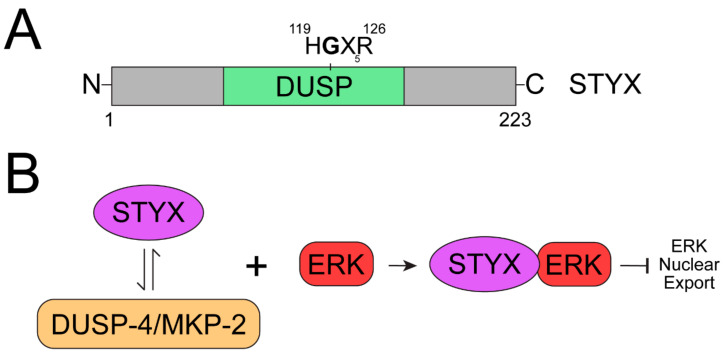
Pseudophosphatase STYX regulates ERK1/2 activity. (**A**) Schematic diagram of the pseudophosphatase STYX. STYX has a DUSP domain, which is catalytically inactive due to a C120G mutation in the active site motif (HCX_5_R). (**B**) Despite lacking catalytic activity and a CH2 domain, STYX competes with DUSP-4/MKP-2 to regulate the nucleocytoplasmic shuttling of ERK1/2. STYX binds ERK and serves as a nuclear anchor, inhibiting the nuclear export of ERK.

**Table 1 ijms-22-12595-t001:** Roles of pseudophosphatases in MAPK signaling and beyond.

Pseudophosphatase	Critical Mutation(s)	Implications in MAPK Signaling	Implications in Other Pathways	Links to Diseases
MK-STYX (STYXL1/DUSP-24)	Contains the active site sequence FSX_5_R instead of HCX_5_R which results in the loss of catalytic activity. The KIM of MK-STYX lacks consecutive arginines (R53V), which may be the reason that MK-STYX does not bind MAPK/ERK.	Does not function like the other MKPs in that MK-STYX does not impact MAPK/ERK signaling.	The overexpression of MK-STYX decreases stress granule formation through interaction with G3BP1 and alters the localization of HDAC6. MK-STYX modulates the activity of PTPM1 in order to induce stress-activated mitochondrial-dependent apoptosis. In PC12 cells, the overexpression of MK-STYX induces neurite formation by affecting cofilin and decreasing RhoA activation.	MK-STYX has a potential oncogenic role in Ewing’s sarcoma family tumors (ESFT) linked to the EWS-FLI1-driven overexpression of MK-STYX in these tumors [38]. Potential oncogenic role in glioblastoma (GBM) as MK-STYX was found to be upregulated and promoted aggressive phenotypes in gliomas [86]. Increased expression of MK-STYX implicated in the proliferation of hepatocellular carcinoma (HCC) by inhibiting apoptosis [87]. Increased expression of MK-STYX noted in breast cancer and prostate cancer [88].
STYX	The essential active site cysteine (C) is replaced by a glycine (G) and results in the loss of catalytic activity.	Competes with MKP-2 to serve as a spatiotemporal regulator of ERK1/2 and reduce downstream MAPK activation. Downregulation of STYX inhibits Golgi polarization in an ERK-dependent manner.	STYX associates with CRHSP-24 (calcium-regulated heat-stable protein of 24 kDa) to serve as a critical regulator of spermatogenesis in mice and the deletion of STYX results in male sterility [89]. STYX regulates ubiquitination by interacting with F-box proteins and inhibiting the associated SCF complex.	The ability of STYX to bind and inhibit the F-box protein FBXW7, along with imbalances in the relative expression of these two proteins, has been implicated in breast cancer, colorectal cancer, and endometrial cancer [72,90,91].
TAB1 (MAP3K7IP1)	The *N*-terminal of TAB1 lacks the three catalytic residues of PP2C (protein phosphatase 2C): -Asp282 (substituted with Glu356 in TAB1), -His62 (substituted with Tyr71 in TAB1), and -Arg33 (missing in TAB1) [76,92]. Four of the active site aspartic acid residues that co-ordinate the metal ions required for catalytic function in PP2C have been substituted: -Asp60 (substituted with Asn69 in TAB1), -Asp239 (substituted with Glu290 in TAB1), -Asp 282 (substituted with Glu356) this is also a catalytic residue, and -Asn283 (substituted with Asp357 in TAB1) [76,92].	Downstream regulator of the MAP3K TAK1 by activating and changing the localization of p38 MAPK in order to recruit p38 MAPK to the TAK1 complex in order to activate TAK1.	Blocks inhibition of p53 by inhibiting the negative regulator MDM2.	TAB1 as part of the TAK1 complex is linked with the outcome of viral infection with enterovirus 71 (EV71), the pathogen responsible for hand, foot, and mouth disease via inhibition of NF-κB activation [93]. TAB1 is implicated as a potential tumor suppressor and lower levels of TAB1 are associated with cancerous ovarian tumors [81]. Implicated in rheumatoid arthritis and other pro-inflammatory diseases [77,78,79].
EGG4/EGG5	Catalytic cysteine (C) residue replaced by aspartic acid (D) in the active site motif [35,36].	Does not regulate MAPK signaling. However, EGG4/5 do regulate dual-specificity tyrosine-regulated kinase (DYRK) signaling. EGG4/5 regulates the oocyte-to-zygote transition in *Caenorhabditis elegans* via competition with mini brain kinase 2 (MBK-2).	EGG4/5 regulates the localization of EGG3 and CHS-1 during meiotic progression and EGG4/5 localize to the cortex in developing oocytes independently of MBK-2.	Loss of function of both EGG4/5 results in maternal-effect lethality in the nematode *C. elegans.*

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
