# Peer review of "Pseudophosphatases as Regulators of MAPK Signaling"

_ijms, 2021, doi:10.3390/ijms222212595_

Round 1
Reviewer 1 Report
The review titled “Pseudophosphatases as regulators of MAPK signaling” by Wilber Hepworth and Hinton discussed a relevant topic for the special edition and is well organized. The authors focused on the contribution of pseudophophosphatases’ roles in MAPK signaling such as MKP-STYX, STYX, and TAB1. However, there are some minor errors in this manuscript:
- Please, there are extra space in several lines: 12, 32, 34, 39, 42, 43, 46, 48, 97, 109, 114, 137, 140, 180, 232, 234, 304. Was there a problem with the formatting of the document?
- In lines 18 and 24: Please, add a space “TAB 1(TAK1- binding protein)...”
- Line 65: The authors may add the abbreviations of: “A three-tiered “core signaling module” consisting of MAP kinase (MAPK), MAP kinase kinase (MAP2K) and MAP kinase kinase kinase (MAP3K)”
- Line 71: “A diverse family of kinases termed MAP kinase kinases or MAP3Ks catalyzes this phosphorylation” Do you mean MAP kinase kinase or MAP kinase kinase kinases (MAP3Ks)?
- Line 83: The authors may change “doesn’t” to “does not”
- Line 123: “In particular, pseudophosphatases are regulators of spermatogenesis, mitogen activated protein kinase (MAPK) signaling, ubiquitylation, demyelination, dual specificity tyrosine-regulated kinases (DYRKs), stress response, apoptosis, RhoA signaling and neuronal differentiation [32, 36]. In addition, pseudophosphatases have been implicated as regulators in MAPK signaling pathways, the focus of this review.”
This paragraph contains repeated information. The authors should remove “mitogen activated protein kinase (MAPK) signaling” and leave the last sentence “In addition, pseudophosphatases have been implicated as regulators in MAPK signaling pathways, the focus of this review”.
- Line 172: Please, add a space “(A)The atypical domains…”
- Please, correct “phosphor-ylate” to “phospho-rylate” and “fig-ure” to “figu-re”
- Line 275: Has the formatting conditions changed? The font size seems less than the rest of the body text.
- Line 288: Please, correct the sentence “They show that that this tight control…”
- The authors should reduce the font size of the Table 1 and figures. It could be better integrated into the text.
- Abbreviations. Please, correct “MAP mitogen-activated protein kinase” that it refers MAPK.
Author Response
Response to Reviewers
Reviewer 1’s Comments
The review titled “Pseudophosphatases as regulators of MAPK signaling” by Wilber Hepworth and Hinton discussed a relevant topic for the special edition and is well organized. The authors focused on the contribution of pseudophosphatases’ roles in MAPK signaling such as MKP-STYX, STYX, and TAB1. However, there are some minor errors in this manuscript:
- Please, there are extra space in several lines: 12, 32, 34, 39, 42, 43, 46, 48, 97, 109, 114, 137, 140, 180, 232, 234, 304. Was there a problem with the formatting of the document?
“We have deleted the extra space.”
- In lines 18 and 24: Please, add a space “TAB 1(TAK1- binding protein)...”
“We have added an extra space.”
- Line 65: The authors may add the abbreviations of: “A three-tiered “core signaling module” consisting of MAP kinase (MAPK), MAP kinase kinase (MAP2K) and MAP kinase kinase kinase (MAP3K)”
“We have provided the acronyms the reviewer suggested.”
- Line 71: “A diverse family of kinases termed MAP kinase kinases or MAP3Ks catalyzes this phosphorylation” Do you mean MAP kinase kinase or MAP kinase kinase kinases (MAP3Ks)?
“We have changed the text to MAP3K.”
- Line 83: The authors may change “doesn’t” to “does not”
“We made the suggested change.”
- Line 123: “In particular, pseudophosphatases are regulators of spermatogenesis, mitogen activated protein kinase (MAPK) signaling, ubiquitylation, demyelination, dual specificity tyrosine-regulated kinases (DYRKs), stress response, apoptosis, RhoA signaling and neuronal differentiation [32, 36]. In addition, pseudophosphatases have been implicated as regulators in MAPK signaling pathways, the focus of this review.”
This paragraph contains repeated information. The authors should remove “mitogen activated protein kinase (MAPK) signaling” and leave the last sentence “In addition, pseudophosphatases have been implicated as regulators in MAPK signaling pathways, the focus of this review”.
“We made the suggested change.”
- Line 172: Please, add a space “(A)The atypical domains…”
“We added the space and changed to “(A) The …” ”
- Please, correct “phosphor-ylate” to “phospho-rylate” and “fig-ure” to “figu-re”
“We made changes that did not require hyphenation, which was caused by reformatting the original work to fit the journal’s requirement.”
- Line 275: Has the formatting conditions changed? The font size seems less than the rest of the body text.
“The formatting was different; it is the corrected 9 font size to designate a figure legend.”
- Line 288: Please, correct the sentence “They show that that this tight control…”
“The correction was made.”
- The authors should reduce the font size of the Table 1 and figures. It could be better integrated into the text.
“The font size was reduced to 9 for figures and table, which is different from 12 in the text. ”
- Abbreviations. Please, correct “MAP mitogen-activated protein kinase” that it refers MAPK.
“We made the correction.”
We are very grateful for the thoroughness and suggestions of this reviewer.
Reviewer 2 Report
The MAPK signaling plays a critical role in many cellular processes and is tightly regulated in cells. Although having no catalytic activity, pseudophosphatases have indispensable role in the regulation of MAPK signaling. In this review manuscript, authors focused on these pseudophosphatases, and summarized the recent proceedings of pseudophosphatase studies in the MAPK signaling, which somehow provides insight for our understanding of the regulation of MAPK signaling. Overall, it is a good review manuscript with some obvious defects. Authors can improve it according to my suggestions as follow:
(1) the major issue for this manuscript is, the summary of literature is not logical and concise. For example, line 52 to line 88, line 158 to line 170. Some information is redundant, such as line 92-93, and line 110. Authors should write those sections systematically.
(2)besides Ser/Thr/Tyr, His can also be phosphorylated by protein kinases, authors may add this to line 42.
Author Response
Response to Reviewers
Reviewer 2’s Comments
The MAPK signaling plays a critical role in many cellular processes and is tightly regulated in cells. Although having no catalytic activity, pseudophosphatases have indispensable role in the regulation of MAPK signaling. In this review manuscript, authors focused on these pseudophosphatases, and summarized the recent proceedings of pseudophosphatase studies in the MAPK signaling, which somehow provides insight for our understanding of the regulation of MAPK signaling. Overall, it is a good review manuscript with some obvious defects. Authors can improve it according to my suggestions as follow:
(1) the major issue for this manuscript is, the summary of literature is not logical and concise. For example, line 52 to line 88, line 158 to line 170. Some information is redundant, such as line 92-93, and line 110. Authors should write those sections systematically.
“It is unclear of what should be rearranged in the sections that this reviewer highlighted. Should the entire sections be we arranged or paragraphs within each section?”
What is meant by systematically? We logically and systematically approach this topic in the following manner to tell a coherent story to new readers.
A. We first discuss MAPK signaling
B. Followed by the phosphatases, MKPs, which dephosphorylates MAPKs
C. Briefly mentioned the catalytically inactive member of MKPs, pseudophosphatase members MK-STYX
D. To introduce readers to pseudophosphatases, we discuss pseudophosphatases
E. Pseudophosphatase MK-STYX is discussed
F. Pseudophosphatases STYX is discussed because it is an important regulator of MAPK signaling
G. Then other pseudophosphatases that have a role in MAPK signaling is discussed
H. Concluded with the importance of pseudophosphatases as regulators of MAPK signaling, which is rarely discussed in the this “old” signaling cascade
Thus, we believe that this is a very logical story to discuss pseudophosphatases role in MAPK signaling. In addition, this systematically highlights that MK-STYX and STYX are not isolated incidents, but demonstrates a wider phenomenon that must be explored.
(2)besides Ser/Thr/Tyr, His can also be phosphorylated by protein kinases, authors may add this to line 42.
Yes, histidine phosphorylation is an exciting post-translational modification that is understudied. We are very interested in histidine phosphorylation and considered adding this to the current text in the following manner.
Histidine phosphorylation is emerging as in important post-translation modification for cellular signaling [1-3].
- Adam, K. & Hunter, T. (2018) Histidine kinases and the missing phosphoproteome from prokaryotes to eukaryotes, Laboratory investigation; a journal of technical methods and pathology. 98, 233-247.
- Adam, K., Lesperance, J., Hunter, T. & Zage, P. E. (2020) The Potential Functional Roles of NME1 Histidine Kinase Activity in Neuroblastoma Pathogenesis, International journal of molecular sciences. 21.
- Fuhs, S. R. & Hunter, T. (2017) pHisphorylation: the emergence of histidine phosphorylation as a reversible regulatory modification, Current opinion in cell biology. 45, 8-16.
However, we decided not to add histidine phosphorylation to this article. Because it is not well understood. In addition, our following sentence is factual and true: “The most common phosphorylation events occur on serine, threonine, and tyrosine residues [4].” Histidine phosphorylation is ~ 6 percent in mammalian cells. Lastly, the currently reports of MAPK signaling involves serine, threonine, and tyrosine residues.
We thank this reviewer for these comments.
Round 2
Reviewer 2 Report
Authors have carried out minor revisions, which does not resolve the major issues in this manuscript, particularly those in two sections I pointed out. For instance, in the first section, there's at least a problem almost in every sentence. Here, I just use the first paragraph as an example, "Mitogen-activated protein kinase (MAPK) signaling pathways are important regula-52 tors in cellular processes such as survival, proliferation, apoptosis, differentiation, and gene transcription." "gene transcription" is not a cellular process, can not be listed together with " survival, proliferation, apoptosis, differentiation", though these processes need "gene transcription". "There are multiple mammalian MAPK pathways, extracellular signal-regulated (ERK1/2; c-JUN NH2-terminus kinase or stress protein activated kinases (JNKs/SAPKs); and p38 MAPK. ERKs are mostly activated by insulin and mitogens; JNK through growth factors, environmental stresses, and proinflammatory stimuli; and p38 MAPKs through stresses, and proinflammatory stimuli." Growth factors such as EGF mainly activate ERK but not JNK. "Thus, MAPKs also have essential roles in stress and inflammation activated pathways, linking them to diabetes and inflammatory disorders." the MAPKs, particularly ERK, are predominantly linked to cancers, which should not be missed. Hence, this manuscript needs to be extensively edited for avoiding any mistakes and redundancies. The current version is still far from an outstanding review.
Author Response
- We removed gene transcription from the sentence as the reviewer suggested.
- We made it clearer what ERK and JNK regulate by stating, “There are multiple mammalian MAPK pathways, including extracellular signal-regulated (ERK1/2); c-JUN NH2-terminus kinase or stress protein activated kinases (JNKs/SAPKs); and p38 MAPK. ERKs are mostly activated by insulin, mitogens, and growth factors, whereas JNK and p38 MAPK are primarily activated through environmental stresses and proinflammatory stimuli. Thus, MAPKs also have essential roles in stress and inflammation.
- We added cancer as a disease that MAPKs control.
- Lastly, we addressed the concern about logic and conciseness for lines 52 to 88 and 158-170, which was made in your first review, by reorganizing and rewording these sections. The marked version demonstrates such changes; we are hopeful that it is visible through reading our manuscript.
We thank this reviewer for these comments. We are hopeful that these revisions address your concerns, and that you continue to consider it an overall good review manuscript.
Round 3
Reviewer 2 Report
No further comments